# Effects of Cardiac Resynchronization Therapy on Cardio-Respiratory Coupling

**DOI:** 10.3390/e23091126

**Published:** 2021-08-30

**Authors:** Nikola N. Radovanović, Siniša U. Pavlović, Goran Milašinović, Mirjana M. Platiša

**Affiliations:** 1Pacemaker Center, University Clinical Center of Serbia, 11000 Belgrade, Serbia; pavlosini@yahoo.com (S.U.P.); goran@milasinovic.com (G.M.); 2Faculty of Medicine, University of Belgrade, 11000 Belgrade, Serbia; 3Institute of Biophysics, Faculty of Medicine, University of Belgrade, 11129 Belgrade, Serbia; mirjana.platisa@med.bg.ac.rs

**Keywords:** heart failure, cardiac resynchronization therapy, cardiopulmonary coupling, heart rhythm, respiratory rhythm, responders, non-responders, sample entropy, cross-sample entropy, detrended fluctuation analysis

## Abstract

In this study, the effect of cardiac resynchronization therapy (CRT) on the relationship between the cardiovascular and respiratory systems in heart failure subjects was examined for the first time. We hypothesized that alterations in cardio-respiratory interactions, after CRT implantation, quantified by signal complexity, could be a marker of a favorable CRT response. Sample entropy and scaling exponents were calculated from synchronously recorded cardiac and respiratory signals 20 min in duration, collected in 47 heart failure patients at rest, before and 9 months after CRT implantation. Further, cross-sample entropy between these signals was calculated. After CRT, all patients had lower heart rate and CRT responders had reduced breathing frequency. Results revealed that higher cardiac rhythm complexity in CRT non-responders was associated with weak correlations of cardiac rhythm at baseline measurement over long scales and over short scales at follow-up recording. Unlike CRT responders, in non-responders, a significant difference in respiratory rhythm complexity between measurements could be consequence of divergent changes in correlation properties of the respiratory signal over short and long scales. Asynchrony between cardiac and respiratory rhythm increased significantly in CRT non-responders during follow-up. Quantification of complexity and synchrony between cardiac and respiratory signals shows significant associations between CRT success and stability of cardio-respiratory coupling.

## 1. Introduction

Heart failure is an increasingly prevalent disease accompanied by significant morbidity and mortality [1]. It is predicted that hospital admissions due to heart failure and the costs of treating these patients will continue to rise in the next decade, [2]. Cardiac resynchronization therapy (CRT) is a well-established therapy for appropriately selected symptomatic patients with heart failure and reduced left ventricular ejection fraction [3]. In most recipients, it slows disease progression, alleviates symptoms, reduces emergency service visits, hospitalizations, and prolongs life [4]. The benefits of cardiac resynchronization therapy have been explained primarily by the increase in the electromechanical synchrony of the ventricles, i.e., by the decrease of conduction delay [5]. However, today we know that CRT implantation leads not only to mechanical and electrical changes but also to changes in the level of neurohormonal activity [6,7,8]. Imbalance of the autonomic nervous system (ANS) is part of the basic pathophysiological characteristic of heart failure [9]. So far, only few studies have examined the ANS modifications that occur in patients after CRT implantation. We know for sure that in all CRT patients, in the first hours after device implantation, an improvement in cardiac neuroadrenergic function is registered, confirmed by increased levels of presynaptic catecholamine uptake, which is a direct acute hemodynamic effect of left and right ventricular synchronization [10,11]. However, the results of chronic follow-up in a small number of patients show that the improvement in sympathetic function, i.e., the increase in catecholamine uptake, is observed exclusively in CRT responders [10]. Moreover, higher levels of catecholamine analogue uptake before CRT implantation are recorded in patients who will become responders to resynchronization therapy, which would mean that the degree of ANS deterioration before CRT implantation determines whether the patient will benefit from device implantation. In other words, responders become those patients in whom the ANS is not structurally impaired but is in a state of hibernation [10].

In heart failure patients, in addition to sinus rhythm, different types of arrhythmias frequently occur, which limits the application of linear methods in the analysis of interbeat interval time series [12]. Therefore, signals should be analyzed by nonlinear techniques, and we applied the sample entropy approach [13] and as a complementary method of detrended fluctuation analysis [14] to changes in cardiac and respiratory rhythm complexity after CRT implantation. These techniques have been demonstrated to provide reliable estimates of signal complexity in cardio-respiratory data [15,16,17].

We hypothesized that there were differences in cardio-respiratory parameters between responders and non-responders to CRT. The aim of this study was to evaluate remote changes in ANS function after CRT implantation, through examination of interactions between cardiac and respiratory signals with complementary measures of time series analysis, before and 9 months after device implantation. We aimed to determine whether and in which direction the observed nonlinear parameters shift and whether their dynamics differed between responders and non-responders to CRT. The second goal of our research is to define the parameters of autonomic function that are capable to preoperatively separate future responders from those who will not benefit from resynchronization therapy. Since 20–40% of patients do not respond to CRT [18], this could contribute to selection of patients with heart failure who are more likely to benefit from this treatment.

## 2. Materials and Methods

### 2.1. Participants and Study Design

This is a one-group, quasi-experimental, pilot study, designed and conducted relying on a template for intervention description and replication checklist and guide [19]. Baseline measurements were performed in 54 patients. In the first year after device implantation, three patients died, two from terminal heart failure and one from massive ischemic stroke. Moreover, follow-up measurements were not performed in four patients because they continued to regularly monitor the function of the device in the regional pacemaker centers. Therefore, we evaluated 47 patients (14 females), mean (SD) age of 64.4 (8.7) years, with heart failure and an indication for CRT device implantation according to current guidelines [3]. Thus, all patients had left ventricular ejection fraction below 35%, left bundle branch block (according to Strauss criteria [20]) on ECG, and all were on optimal therapy for heart failure (Table 1). St. Jude Medical CRT devices were implanted in all patients (St. Jude Medical Allure, St. Jude Medical Quadra Allure MP, St. Jude Medical Quadra Assura MP). The study was approved by the Ethics Committee of the Faculty of Medicine the University of Belgrade, and each subject signed an informed consent (approval date: 17 March 2017, Ref. Numb.29/III-4). Measurements were performed before (baseline) and approximately 9 months after CRT device implantation (follow-up). After follow-up, patients were divided into two groups, responders (*N* = 27) and non-responders (*N* = 20), in relation to the response to CRT, which was assessed according to changes in certain clinical and echocardiographic parameters. More precisely, we defined that an echocardiographic responder to CRT was a patient who increased LVEF by 5% or more at follow-up, while clinical response was based on NYHA classification and a six-minute walk test result and was considered positive if NYHA improvement was ≥1 class and distance covered during the test increased by 10%. In this study, the CRT responder had to meet the stated echocardiographic and clinical criteria.

### 2.2. Signals Measurements and Processing

Both experiments (baseline and follow-up) were conducted in the morning, between 7:00 and 8:00 a.m. in a quiet surrounding at the Pacemaker Center of the University Clinical Center of Serbia. Baseline measurements were done immediately before device implantation. For follow-up recording, CRT device was switched off, i.e., spontaneous rhythm was used for cardiac signal examination. Data were acquired from 20 min of ECG (RR intervals) and respiratory signal measurement of relaxed subjects in supine position, breathing spontaneously, with a sampling frequency of 1000 Hz by the Biopac MP100 system and AcqKnowledge 3.9.1 software (BIOPAC System, Inc., Santa Barbara, CA, USA). The ECG data were collected using the ECG 100C electrocardiogram amplifier module. The RSP 100C respiratory pneumogram amplifier module with TSD 201 transducer attached to the belt (adjustable nylon strap) was used to measure abdominal expansion and contraction [21]. The interbeat interval time series in patients with sinus rhythm and sinus rhythm with ventricular extrasystoles and with atrial fibrillation were analyzed. Interbeat (RR) intervals and interbreath (BB) intervals were extracted from recorded signals using tool Pick Peaks from OriginPro 8.6 (OriginLab Corporation, Northampton, MA, USA). Heart rate (HR) and breathing frequency (BF) were obtained as a reciprocal value of subject mean RR interval and mean BB interval. The resampling procedures were performed to obtain equal equidistant resampled time series (time series with the same number of equidistant samples) of RR intervals and respiratory signals [15].

### 2.3. Sample Entropy and Cross Sample Entropy

Sample entropy (SampEn) and cross-sample entropy (crossSampEn) are non-linear measures of time series regularity derived from the probability of finding a similar pattern within signal/signals [13]. It is defined as the negative natural logarithm of the conditional probability that two sequences similar for *m* points remain similar within tolerance *r* at the next point, where self-matches are not included. We calculated SampEn from RR interval series (SampEnRR) and corresponding respiratory signal (SampEnResp). The signal with a small number of similar patterns is characterized by higher values of SampEn which indicates its greater unpredictability (irregularity). Similarly, coupling of two signals with a small number of similar patterns would result in high values of crossSampEn indicating their high asynchrony i.e., low association between analyzed systems. SampEn and crossSampEn were calculated with fixed input variables *m* = 2 (window size) and *r* = 0.15·SD (tolerance, SD—standard deviation of time series) from whole time series (approximately about 1200 samples). Previously developed MATLAB scripts were used to calculate SampEn and crossSampEn from normalized and two equally equidistant resampled series of RR intervals and of the respiratory signal [15]. Entropy measures and scaling exponents were computed using MATLAB (R2007b, TheMathWorks, Inc., Natick, MA, USA).

### 2.4. Detrended Fluctuation Analysis (DFA)

Detrended fluctuation analysis was used to calculate short-term *α*_1_ and long-term *α*_2_ of the equally equidistant resampled series of RR intervals and respiratory signal data [14,20]. In general, for the time series *x*(*i*), of length *N,* one calculates the profile *y*(*k*):(1)$$y\left(k\right)=\sum_{i=1}^{k}\left[x_{i}-\bar{x}\right]$$

The profile *y*(*k*) is divided into non-overlapping segments of length *n*. In each segment, the local trend is the linear least squares fit for the segment data. The squared fluctuation of the segment is calculated from detrended walk which is the difference between *y*(*k*) and the local trend *y_n_*(*k*). The root-mean-square fluctuation function of the time series is determined by averaging over all segments:
(2)$$F\left(n\right)=\sqrt{\frac{1}{N}\sum_{k=1}^{N}{\left[y\left(k\right)-y_{n\left(k\right)}\right]}^{2}}$$

The scaling exponent is defined as the slope of the linear regression line of log(*F*(*n*)) against log(*n*). We found one crossover, i.e., two different scaling exponents for both time series at *n* = 16. The value of scaling exponents indicates correlation properties: *α* < 0.5 (anticorrelated), *α* ~ 0.5 (uncorrelated—white noise), *α* ~ 1 (correlated—1/*f* noise), and *α* ~ 1.5 (strongly correlated—Brownian noise). Detrended fluctuation analysis typically shows two ranges of scale invariance, which are quantified with two separate scaling exponents, *α*_1_ and *α*_2_, reflecting the short-term and long-term correlations, respectively. Previously developed MATLAB scripts were used to calculate scaling exponents from normalized and two equally equidistant resampled series of RR intervals and of the respiratory signal [22].

### 2.5. Statistical Analysis

At the beginning of the study, we calculated the sample size with a population of 54 patients, a confidence level of 95% and 5% of confidence interval. Using these parameters, we found that the estimated recommended sample size was 47 patients.

Mixed design ANOVA for repeated measures was applied to find significant group x measures interaction. Separately, because we analyzed two groups of patients (responders and non-responders to CRT) with repeated measurements, we used one-way ANOVA for intergroup comparisons at the baseline and at the follow-up measures. On the other side, we used repeated measures ANOVA for intragroup comparisons between baseline and follow-up data. The results of intergroup comparisons are presented in Table 2, and statistically significant results of intra- and intergroup comparisons are presented in Figures.

We estimated effect sizes, as a measure of magnitude of the difference between responders and non-responders groups by Cohen’s term *d*_s_ for all calculated parameters and presented the results in Table 2. In computing, we used pooled standard deviation for both groups of parameters obtained in follow-up measurement. Cohen classified effect sizes as small (*d* = 0.2), medium (*d* = 0.5) and large (*d* ≥ 0.8) [23].

We applied ROC analysis to examine the diagnostic ability of the ratio of CrossSampEn value after and before CRT to discriminate patients with successful resynchronization therapy. We calculated the area under the ROC curve that can be used as a criterion for measuring the discriminative ability of variables in a given clinical situation.

Statistical analyses were performed using the software package SPSS Statistics (version 17.0, SPSS Inc., Chicago, IL, USA). For all analyses, probability values less than or equal to 0.05 were considered as statistically significant.

## 3. Results

Table 1 shows demographic and clinically relevant, baseline data of CRT patients. Baseline clinical parameters did not differ significantly between groups of future CRT responders and non-responders, except that there were significantly more patients with ischemic heart disease among patients who would not benefit from resynchronization therapy (Table 1).

Clinical and echocardiographic examination before follow-up signal recording showed a statistically significant difference in left ventricular ejection fraction, NYHA class, and distance covered on a six-minute walk test between CRT responders and non-responders (Table 2).

We found that in all patients, resting heart rate decreased significantly after CRT implantation (Figure 1), but the complexity of the RR intervals, measured by SampEnRR, did not change. However, in CRT non-responders SampEnRR was higher in both the baseline and follow-up measurements, but the differences were not statistically significant (Table 2, Figure 1).

By additional complementary measures, short-term, *α*_1_(RR), and long-term, *α*_2_(RR), scaling exponents of RR intervals, some of these differences were statistically determined (Figure 2). Namely, both scaling exponents were higher in CRT responders, which indicated a more correlated RR interval time series or more regular time series than in non-responders. Statistically significant differences between groups were achieved for *α*_1_(RR) at follow up measurements and for *α*_2_(RR) at baseline measurements (Figure 2). The long-term scaling exponent of RR interval series was the only parameter that could preoperatively separate future responders from those who would not benefit from resynchronization therapy (Figure 2).

There was no significant difference between CRT responders and non-responders in breathing frequency at the baseline measurement (Figure 1). Compared to the baseline recording, at the follow-up examination in responders to CRT, respiratory rate was significantly lower, while in non-responders, it was slightly higher. These findings resulted in a statistically significant difference in breathing frequency between responders and non-responders after follow-up measurements (Figure 1). In CRT non-responders, respiratory rhythm complexity quantified by sample entropy, SampEnResp, at baseline was significantly lower than after CRT implantation and slightly lower than in responders at both measurements (Figure 1). Respiratory short-term, *α*_1_(Resp), and long-term, *α*_2_(Resp), scaling exponents had the same direction of changes in all patients with a statistically significant increase in CRT responders and only for *α*_2_(Resp) in non-responders (Figure 2). Exceptional behavior at follow-up measurements was found for *α*_1_(Resp) in CRT non-responders, where there was no increase in the correlation of respiratory time series, as in other cases after CRT implantation. These findings probably reflected the absence of some respiratory control mechanisms over short scales in CRT non-responders that may differentiate responders from non-responders.

In Figure 3, we presented the changes of CrossSampEn between RR interval and respiratory signal time series in this study. In CRT responders, CrossSampEn did not change significantly between baseline and follow-up measurements. Contrary, in non-responders, after CRT implantation, CrossSampEn was statistically significantly reduced compared to the baseline examination, achieving values that were statistically significantly higher than in CRT responders (Figure 3).

Further, we calculated the ROC curve values for the ratio (rCrossSampEn) defined as the quotient of CrossSampEn obtained for follow-up and baseline measurements (Figure 4). According to the characterization of ROC curves, our result of the area under the ROC curve belongs to the fair category and rCrossSampEn was not as strong a discriminator as we expected, but this result may indicate the direction for application of similar parameters in further studies.

It is important to emphasize that the underlying cardiac rhythm did not change in any of observed patients at the follow-up compared to the baseline examination.

## 4. Discussion

To our knowledge, this is the first study designed and conducted to determine the effect of cardiac resynchronization therapy on the relationship between the cardiovascular and respiratory systems in patients with heart failure. In this research, we used available methods that have been shown to provide the best assessment of cardiac and respiratory signal complexity.

In CRT responders, 9 months after device implantation, the resting heart rate is significantly reduced. This is the result of restoring autonomic balance, primarily the recovery of vagal activity that is predominantly responsible for resting heart rate control [24]. This finding confirms that the clinical progress in patients with heart failure, i.e., NYHA class improvement, is associated with a lower resting rate [24]. Moreover, after CRT implantation, antiarrhythmics are more freely prescribed because there is no longer a fear of bradycardia developing, and in patients with atrial fibrillation as the underlying rhythm, higher doses of these drugs are necessary to achieve the target percentage of biventricular pacing. More intensive antiarrhythmic therapy is actually the main reason why CRT non-responders also had lower resting ventricular rate at follow-up compared to baseline examination. Moreover, all patients were one year older at the follow-up, and we know that with each year of life the heart rate decreases [25]. An indicator of better autonomic control and baroreflex activity in patients who had a favorable response to resynchronization therapy is that they have a higher complexity of RR intervals, i.e., a more irregular heart rhythm on the follow-up. Indeed, baseline and follow-up values of SampEnRR do not differ statistically significantly, but it is clear that the values of this parameter in CRT responders and non-responders move in different directions on the control recording. Certainly, the optimization of therapy for heart failure after CRT implantation also plays an important role, as there is evidence that ß-blockers, some ACE inhibitors, and aldosterone antagonists lead to an increase in vagal tone [26,27,28]. Therefore, the improvement in cardiac neuroadrenergic, parasympathetic, and baroreflex function due to cardiac resynchronization and optimization of drug therapy leads to an increase in heart rate variability and heart rate turbulence, which results in more irregular heart rhythm with lower resting rate.

In CRT responders, a significantly lower respiratory rate was registered at the control examination compared to the baseline recording, which resulted in a statistically significant difference in the values of this parameter between responders and non-responders after follow-up measurements. Respiratory rate directly depends on the extent of stimulation of peripheral and central chemoreceptors. In CRT responders, cardiac systolic function recovers; oxygen delivery to the periphery increases, i.e., better blood flow is achieved in organs, especially in the kidneys, which reduces hypoxia and metabolic acidosis, resulting in reduced chemoreceptor stimulation and reduced respiratory rate [29,30,31]. In heart failure, there is an increased sensitivity of peripheral chemoreceptors to a drop in partial pressure of oxygen, and it is assumed that their sensitivity decreases with the improvement of cardiac function [32]. In CRT responders, it is also likely (at least in part) that there will be a reduction in respiratory muscle weakness. This would mean a decrease in the production of specific metabolites, which leads to a decrease in the activity of muscle parenchyma chemoreceptors, i.e., a muscle metaboreflex activity reduction, which contributes to a decrease in sympathetic tone, respiratory rate, dyspnea and improved aerobic function [33,34]. In future studies, it would be interesting to examine the joint contribution of resynchronization therapy and inspiratory muscle training, which we already know reduces the respiratory muscle metaboreflex. A significant reduction in respiratory rate during follow-up in CRT responders is a confirmation of the previously shown relationship of respiratory rate with NYHA class and left ventricular ejection fraction [35]. Thus, in patients who have benefited from CRT implantation, i.e., in whom there was a decrease in the NYHA class and an increase in the left ventricular ejection fraction, a reduction in breathing frequency was registered. In previous research, we have shown that in the presence of impaired autonomic control, a more regular heart rhythm is related to a higher respiratory rate [35]. In the present study, in patients who are non-responders to resynchronization therapy, with a pronounced imbalanced autonomic nervous system, we found a reduction in complexity of heart rhythm and an increase in breathing frequency at the follow-up examination.

It is interesting that in CRT responders the respiratory rhythm was more irregular at the follow-up compared to baseline recording, with no statistical significance, and in CRT, non-responders statistically significantly more irregular. In addition, patients who would have a favorable response to resynchronization therapy had higher values of SampEnResp before device implantation, and this was on the edge of statistical significance. It is difficult to analyze the regularity of respiratory rhythm in patients with heart failure, especially based on a relatively short record during wakefulness, because it is influenced by many factors, such as age, sex, left ventricular ejection fraction, NYHA class, and presence of breathing disorders [36,37]. It has been shown that the frequency of respiratory disorders is lower in women and patients on beta-blocker therapy, while the prevalence of these disorders during sleep and wakefulness increases with deteriorating in cardiac function [36,37]. In previous research, we have concluded that among patients with heart failure, those who have atrial fibrillation as the underlying rhythm have the highest values of SampEnResp [17]. In the same study, we have concluded that respiratory rhythm is more regular in patients with heart failure and without atrial fibrillation than in healthy controls [17]. Thus, the fact that, prior to device implantation, future CRT responders compared to non-responders have significantly higher values of SampEnResp probably indicates that they have a lower degree of structural impairment of ANS, as well as that they still have preserved compensatory mechanisms, which have not yet been sufficiently examined and understood. The increase in respiratory rhythm irregularity at follow-up in patients who did not respond favorably to CRT can be explained by the more frequent occurrence of respiratory disorders in these patients, due to further deterioration of cardiac function but also by the fact that male and patients with atrial fibrillation were more often CRT non-responders in our study.

Synchronization of cardiac and respiratory rhythms slightly increased after device implantation in CRT responders, while in non-responders it was statistically significantly reduced compared to the baseline examination, achieving values of CrossSampEn that were statistically significantly higher in these patients than in CRT responders. Data on cardiopulmonary synchronization in heart failure are deficient in the literature. Furthermore, unlike respiratory sinus arrhythmia, for this type of cardio-respiratory interaction, we do not know enough about underlying physiologic mechanism nor about its significance even in healthy subjects [38,39]. Cardiopulmonary synchronization is considered to be of great importance for fine matching of ventilation and perfusion and thus maximizing energy utilization [39]. It has been shown that synchronization of cardiac and respiratory rhythms in swimmers and athletes is superior to that in ordinary people, i.e., that cardiopulmonary enhancement represents a better physiological state [38,40]. Cardiac and respiratory rhythms are directly controlled by the ANS, but also with influences from the central nervous system [39]. In our previous study, we concluded that cardiopulmonary synchronization, despite the existence of ANS imbalance, is maintained in patients with heart failure who are in sinus rhythm, probably due to powerful effect of compensatory mechanisms [17]. However, the results of this study indicated that with the progression of heart failure, with a further increase in sympathetic activity, the compensatory mechanisms lose their importance. In other words, with further deterioration of autonomic control, the respiratory system is no longer capable to adapt to changes in heart rhythm, leading to a loss of synchrony in the cardio and respiratory coupled systems.

The autonomic nervous system, more precisely baroreflex activity, is responsible for the short-term regulation of heart rate and blood pressure [41]. Findings from previous studies have indicated a strong negative correlation of the short-term scaling exponent of RR interval series with the level of sympathetic activity, i.e., with the plasma levels of circulating catecholamines [42]. Therefore, it is expected that in heart failure, accompanied by strong sympathetic activation, a decrease in the value of this exponent is recorded [43]. Moreover, it has been shown that lower *α*_1_(RR) is associated with increased mortality and hospitalization rate in patients with heart failure and that this exponent is independent predictor of heart failure [43,44]. We found that in CRT responders, nine months after device implantation, *α*_1_(RR) increases slightly but reaches values that are statistically significantly higher than in non-responders. Therefore, this difference in *α*_1_(RR) between responders and non-responders is primarily due to a further decrease in *α*_1_(RR) in patients in whom the deterioration of cardiac function was not stopped, rather than an increase in the value of this parameter in CRT responders. CRT restores autonomic balance leading to ß-adrenergic signaling depression. However, recent findings from animal models and few clinical studies have suggested that restoring sympathovagal balance also involves modulation of cholinergic activity [45]. This explains the statistically insignificant increase in *α*_1_(RR) in CRT responders in our study. Restoration of ANS balance in these patients implies a reduction of sympathetic activity, which leads to an increase in *α*_1_(RR) but also an improvement in vagal tone, which is associated with a decrease in the value of this parameter [46]. In CRT non-responders, a further increase in sympathetic and withdrawal in vagal activity, with an increase in respiratory rate which additionally reduces *α*_1_(RR), leads to a significant decrease in the value of this exponent at the follow-up examination [47,48]. Knowing that *α*_1_(RR) is strongly associated with the breathing pattern, obtained results suggest that CRT non-responders developed alterations in cardiac rhythm related to respiration during follow-up.

The long-term scaling exponent of RR interval series has been less studied in heart failure, but the results of several studies have shown that its values are statistically significantly higher in patients with heart failure with reduced ejection fraction compared to healthy controls [43,49,50]. In this work, we found a slight reduction of *α*_2_(RR) at follow-up examination in CRT responders. It is not a surprising result, because we know that a decrease in sympathetic tone leads to a decrease in *α*_2_(RR) and that the values of this parameter in patients with heart failure with preserved ejection fraction (many CRT responders become part of that subpopulation) do not differ significantly compared to healthy subjects [41,46]. However, we expected statistically significant changes in the value of this exponent both in responders and non-responders, especially since this is the only parameter that separated these two groups of patients at the baseline examination. Our results indicate that *α*_2_ RR should not be considered only as an indicator of the state of ANS, i.e., the degree of blockade of the ß receptors but also as a marker of preservation of the function of the sinoatrial node cells [51]. In our study, in CRT non-responders, *α*_2_(RR) practically does not change, so the values of this parameter between responders and non-responders at the follow-up do not differ significantly due to the decrease in *α*_2_(RR) in CRT responders. Therefore, lower *α*_2_(RR) in patients with heart failure should be definitely understood both as an index of reduced sympathetic activity and as a sign of sinus node dysfunction. This would also mean that the success of resynchronization therapy in conditions of impaired function of sinoatrial node cells is limited. Of course, the baseline values of *α*_2_(RR), as well as the dynamics of this parameter during follow-up, should also be analyzed from the aspect of preserving some slower physiological regulatory mechanisms, probably neuroautonomic, which have not yet been fully examined [50].

Fractal organization of breathing patterns in adults is much less examined, and in pathological conditions, it is limited almost exclusively to respiratory diseases [52,53]. Certainly, the respiratory time series also have nonstationary characteristics, and nonlinear methods, such as detrended fluctuation analysis, are needed to quantify their scaling behavior. So far, we know that long-range correlations are present in human respiratory dynamics and that can be reduced in elderly men [52]. It is considered that in physiological conditions, there is a statistically significant difference between values of short- and long-term scaling exponent of respiratory signal series, with dominance of *α*_1_(Resp) and values of *α*_2_(Resp) in the range of anticorrelations, which indicates the coordinated activity of multiple regulatory mechanisms of the respiratory rhythm [22]. Raoufy et al. have shown that in asthmatic patients compared to healthy controls, long-range correlations are significantly decreased and that nonlinear analysis of respiratory rate variability in these patients has a diagnostic value and can be useful in differentiating uncontrolled from controlled and non-atopic from atopic asthma [53]. In our study, at baseline examination, future CRT responders had almost identical values of *α*_1_(Resp) and *α*_2_(Resp), and in non-responders, long-term scaling exponent had slightly higher values. During follow-up, there was a statistically significant increase in the values of both *α*_1_(Resp) and *α*_2_(Resp) in CRT responders, while in non-responders with a significant increase in *α*_2_(Resp), a significant decrease in *α*_1_(Resp) was registered 9 months after device implantation. An explanation of the dynamics of these scaling exponents in patients who responded unfavorably to resynchronization therapy should probably be sought in the augmented sympathetic activity, reduced vagal tone and increased respiratory rate in these patients. In CRT responders, despite the increase in *α*_1_(Resp), the values of this exponent remain lower than *α*_2_(Resp) at the follow-up examination. This result is an indication that we still do not know enough about the interactions over multiple time scales of different control and feedback systems but also that the fact that the patient has responded favorably to resynchronization therapy does not mean that he no longer has heart failure, i.e., that the ANS imbalance is no longer present [52].

During follow-up, from control to control, there were changes in cardiovascular medications, in relation to the achieved improvement or deterioration of cardiac function, the percentage of biventricular pacing, and the occurrence of atrial and ventricular arrhythmias. However, these changes were mainly related to the doses of administered drugs, rather than to the introduction of new groups of drugs. The effect of these changes, primarily from beta blockers (which have been shown to not only reduce sympathetic activity but also increase vagal tone), on the examined parameters of cardiac and respiratory signal complexity was not analyzed, and this is a limitation of our study.

## 5. Conclusions

In this study, the complexity and synchrony of cardiac and respiratory signals after cardiac resynchronization therapy were examined, and a significant association between CRT success and cardio-respiratory coupling stability was demonstrated. We showed that the entropy values of the RR intervals were higher, although statistically insignificantly, in CRT non-responders both before and after device implantation. Both scaling exponents of RR interval series were higher in CRT responders at the baseline examination, and *α*_2_(RR) was the only parameter that could preoperatively separate future responders from those who would not benefit from resynchronization therapy. At the follow-up recording, in responders to CRT, *α*_1_(RR) increased slightly but reached values that statistically are significantly higher than in non-responders. Respiratory scaling exponents increased at follow-up in both groups, with the exception of *α*_1_(Resp) whose values were significantly reduced in CRT non-responders. Synchronization of cardiac and respiratory rhythms was preserved after device implantation only in CRT responders.

## Figures and Tables

**Figure 1 entropy-23-01126-f001:**
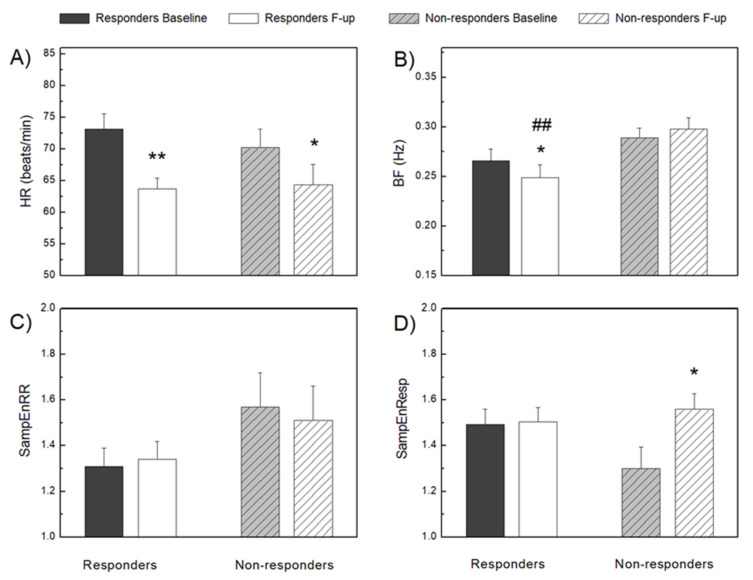
Cardiac and respiratory signal analysis parameters in CRT responders and non-responders, before (baseline) and after (follow-up) device implantation. Heart rate—HR (**A**), breathing frequency—BF (**B**), sample entropy of RR interval series—SampEnRR (**C**), and sample entropy of respiratory signal time series—SampEnResp (**D**). ** *p* < 0.01, * *p* < 0.05 intragroup comparison (F-up vs. Baseline), ## *p* < 0.01 intergroup comparison (Responders F-up vs. Non-responders F-up). Values are mean + standard error.

**Figure 2 entropy-23-01126-f002:**
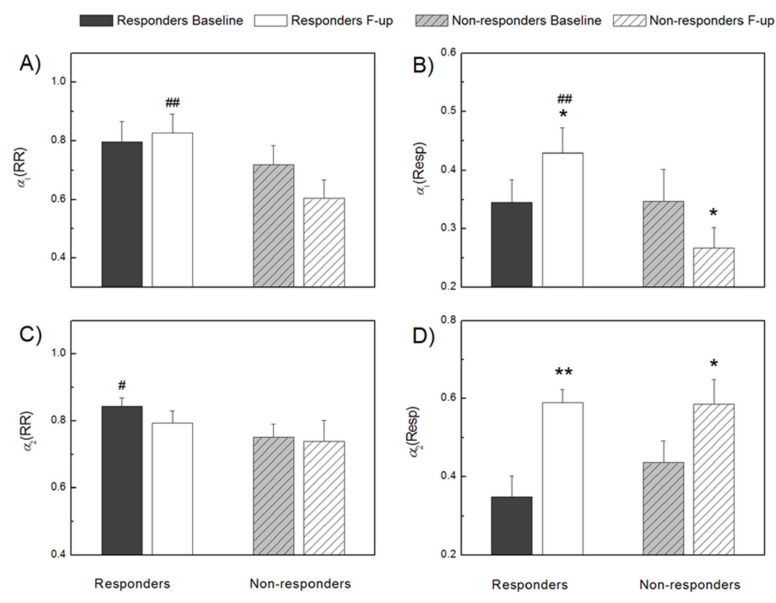
Cardiac and respiratory signal parameters obtained by detrended fluctuation analysis in CRT responders and non-responders, before (baseline) and after (follow-up) device implantation. Short-term scaling exponent of RR interval series—*α*_1_(RR) (**A**), short-term scaling exponent of respiratory signal series—*α*_1_(Resp) (**B**), long-term scaling exponent of RR interval series—*α*_2_(RR) (**C**), and long-term scaling exponent of respiratory signal series—*α*_2_(Resp) (**D**). ** *p* < 0.01, * *p* < 0.05 intragroup comparisons (F-up vs. Baseline), ## *p* < 0.01 Responders F-up vs. Non-responders F-up, # *p* < 0.05 Responders Baseline vs. Non-responders Baseline, intergroup comparisons. Values are mean + standard error.

**Figure 3 entropy-23-01126-f003:**
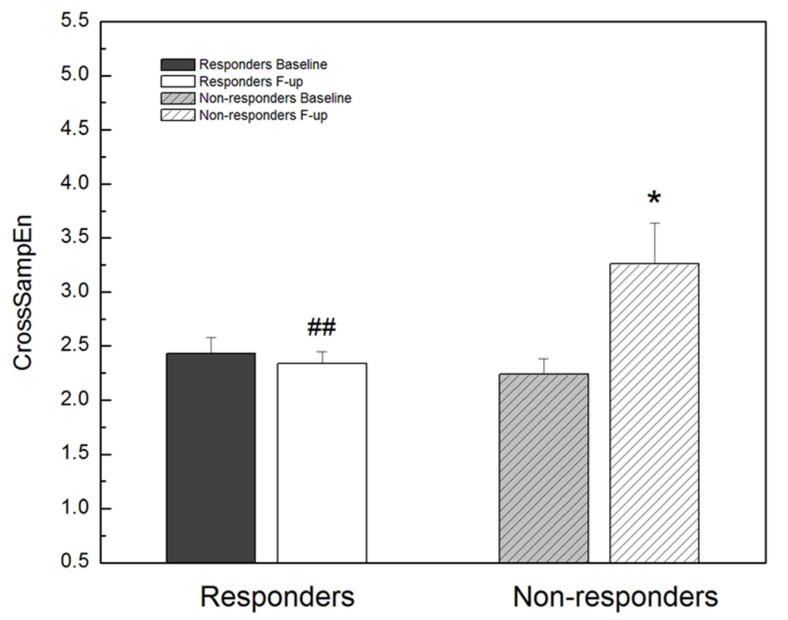
Cross sample entropy between RR interval series and respiratory signal series (CrossSampEn) in CRT responders and non-responders, before (baseline) and after (follow-up) device implantation. * *p* < 0.05 F-up vs. Baseline, ## *p* < 0.01 Responders F-up vs. Non-responders F-up. Values are mean + standard error.

**Figure 4 entropy-23-01126-f004:**
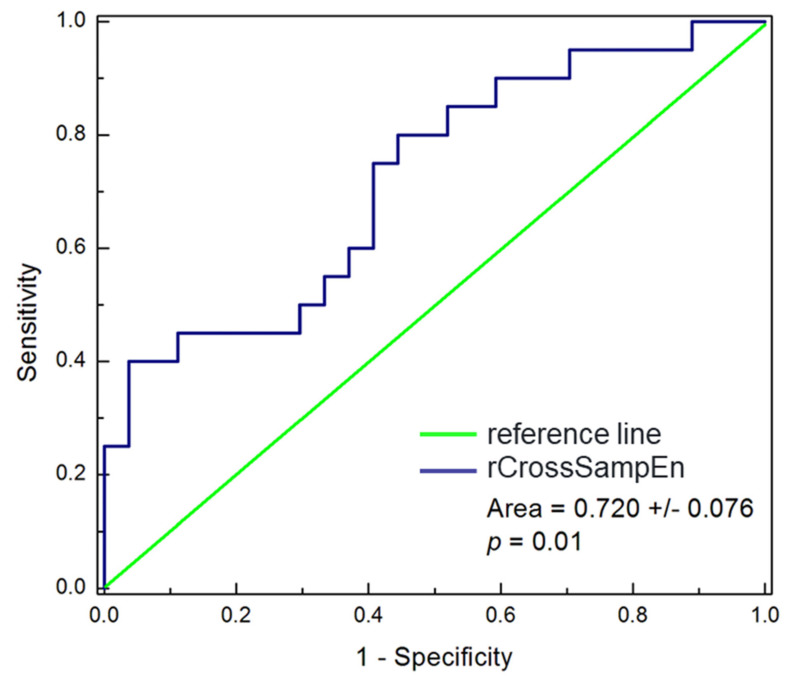
ROC curve for the ratio of cross sample entropy obtained after and before CRT. rCrossSampEn = (CrossSampEn)_F-up_/(CrossSampEn)_Baseline_.

**Table 1 entropy-23-01126-t001:** Anthropometric data and baseline clinical parameters in CRT patients.

Parameter	CRT Responders (%)	CRT Non-Responders (%)	*p*
Age (years)	62.8 ± 8.8	66.5 ± 8.3	0.20
Sex (male)	17 (62.9)	16 (80.0)	0.21
Atrial fibrillation	6 (22.2)	8 (40.0)	0.19
Ischemic heart disease	5 (18.5)	10 (50.0)	0.02
Arterial hypertension	21 (77.8)	17 (85.0)	0.53
Diabetes	7 (25.9)	4 (20.0)	0.64
Hyperlipidemia	14 (51.9)	12 (60.0)	0.58
Tobacco smoking	12 (44.4)	13 (65.0)	0.16
COPD	4 (14.8)	5 (25.0)	0.38

Chronic obstructive pulmonary disease—COPD.

**Table 2 entropy-23-01126-t002:** Clinical parameters and parameters of analysis of cardiac, respiratory rhythm and cardio-respiratory interactions with comparison between responders and non-responders to cardiac resynchronization therapy (CRT) from measurements before (baseline) and after (follow-up) device implantation.

	Baseline			Follow-Up			
	Responders (*N* = 27)	Non-Responders (*N* = 20)	*p*	Responders (*N* = 27)	Non-Responders (*N* = 20)	*p*	Cohen’s *d_s_*
BiVP (%)				96 ± 0.7	95 ± 1	0.85	0.25
LVEF (%)	25 ± 2	26 ± 2	0.38	43 ± 2	24 ± 1	0.04	2.01
NYHA	2.4 ± 0.1	2.6 ± 0.1	0.19	1.6 ± 0.1	2.6 ± 0.1	0.01	1.67
6MWT	280 ± 10	230 ± 20	0.34	360 ± 20	230 ± 20	0.01	1.92
HR (beat/min)	73 ± 2	70 ± 3	0.44	64 ± 2	64 ± 3	0.84	0.06
BF (Hz)	0.27 ± 0.01	0.29 ± 0.01	0.17	0.25 ± 0.02	0.30 ± 0.01	0.01	0.14
SampEnRR	1.31 ± 0.08	1.6 ± 0.2	0.10	1.34 ± 0.08	1.5 ± 0.2	0.30	0.31
SampEnResp	1.49 ± 0.07	1.30 ± 0.09	0.09	1.50 ± 0.06	1.56 ± 0.07	0.55	0.18
CrossSampEn	2.4 ± 0.2	2.2 ± 0.1	0.35	2.3 ± 0.1	3.3 ± 0.4	0.01	0.76
*α*_1_(RR)	0.80 ± 0.07	0.72 ± 0.07	0.43	0.83 ± 0.07	0.60 ± 0.07	0.01	0.77
*α*_2_(RR)	0.84 ± 0.03	0.75 ± 0.04	0.04	0.79 ± 0.04	0.74 ± 0.06	0.43	0.23
*α*_1_(Resp)	0.35 ± 0.04	0.35 ± 0.06	0.97	0.43± 0.04	0.27 ± 0.04	0.01	0.83
*α*_2_(Resp)	0.35 ± 0.05	0.44 ± 0.06	0.26	0.59 ± 0.03	0.59 ± 0.06	0.96	0.16

Biventricular pacing—BiVP, Left ventricular ejection fraction—LVEF, six-minute walk test—6MWT, heart rate—HR, breathing frequency—BF, sample entropy of RR interval series—SampEnRR, sample entropy of respiratory signal time series—SampEnResp, cross-sample entropy—CrossSampEn, short-term scaling exponent of RR interval series—*α*_1_(RR), long-term scaling exponent of RR interval series—*α*_2_(RR), short-term scaling exponent of respiratory signal series—*α*_1_(Resp), long-term scaling exponent of respiratory signal series—α_2_(Resp). *p*—significance test result for intergroup comparisons (responders vs. non-responders). Cohen’s *d_s_*—measure of effect size. Values are mean ± standard error.

## Data Availability

The data presented in this study are available on request from the corresponding author.

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
