# Peer review of "Effects of Cardiac Resynchronization Therapy on Cardio-Respiratory Coupling"

_entropy, 2021, doi:10.3390/e23091126_

Round 1

Reviewer 2 Report

In their manuscript “Effects of cardiac resynchronization therapy on cardio-respiratory coupling”, Nikola N. Radovanovic et al evaluated cardio-respiratory coupling in 47 heart failure patients before and after CRT implantation. To account for changes in cardiac rhythm (e.g., atrial fibrillation or premature beats), they used the sample entropy approach instead of linear methods to associate changes in heart rate with the respiratory cycle. Nine months after implantation, all patients had lower resting heart rate, while reduced breathing rate was only seen in CRT responders. Furthermore, in non-responders a higher asynchrony between cardiac and respiratory cycles was found.

In my opinion, this is a quite important study as it identifies entropy analysis as useful tool to discover effects of CRT therapy on cardiac and respiratory cycles. However, there are currently major flaws in data reporting and the methodology of the study:

  1. Baseline data of patients undergoing CRT are completely missing, for example: NYHA status, ECG characteristics (LBBB or RBBB), QRS duration, ischemic aetiology, gender, comorbidities, medication status, left ventricular ejection fraction, CRT brand. Furthermore, NYHA, LVEF, QRS duration etc at follow up are missing.
  2. There is no detailed definition of “CRT response”. The authors only state “response was assessed according to changes in certain clinical and echocardiographic parameters”. This definition is unacceptable for a clinical study. The authors should give exact thresholds (e.g. in NYHA status or LVEF change). If there was actually no exact threshold, a post-hoc analysis with exact thresholds would be very interesting.
  3. There may be an effect of CRT pacing itself on cardiac cycles, especially in the presence of cardiac arrhythmias. If no sinus rhythm is present (for example in atrial fibrillation), the CRT will try to pace as high as possible, aiming for a pacing rate of > 99%. In these patients, CRT directly controls the heart rate, not the patient. In my opinion, such patients may not be included in such a trial. Please report the proportion of patients with AF or other arrhythmias in your cohort.
  4. Please account for changes in medication, as cardiovascular medication may have a major impact on heart rate and cardio-respiratory coupling.

Minor comments:

  • Please rephrase the last sentence of the first paragraph at page 2 (too complicated).
  • Please provide % pacing in responders vs. non-responders.
  • Was there a loss of follow up? Please provide duration of actual follow up.

Reviewer 3 Report

Thanks for the opportunity to review this interesting manuscript.  Nevertheless, some issues should be clarified:

-A clear hypothesis should be added at the end of the introduction.

-A clear study design with an appropiate followed guidelines should be added at the beginning of the methods section in a subsection called study design. As a quasi-experimental study, the TIDieR guinelines should be cited and this criteria should be named in the study desing subsection. A trial registry should be provided (please, see ClinicalTrials.gov).

-The sample size calculation is lacking and should be added and clearly detailed.

-The statistical analysis needs futher clarification. Please, expand this section, normality and post-hoc analyses should be stated. ROC curve should be explained and detailed. Effect sizes should be added.

-Table 1 should analyzed with a 2x2 ANOVA or analyzing the difference between both groups.

-In results, effect sizes should be added.

-Discussion, in futures studies may be useful to discuss about inspiratory muscle training in addition to your proposed intervention. Please, see:

J. Clin. Med. 20209(6), 1710; https://doi.org/10.3390/jcm9061710

Int. J. Environ. Res. Public Health 202118(4), 1697; https://doi.org/10.3390/ijerph18041697

Round 2

Reviewer 1 Report

The manuscript is markedly improved. Attention to the following would make it even better.

Minor comments:

  1. On page 2, first paragraph, change: “Also, higher level of catecholamine analogue uptake…” to “Also, higher levels of catecholamine analogue uptake…”.
  2. On page 2, third paragraph, change: “The aim of this study is…” to “The aim of this study was…”. Likewise, change “aim” to “aimed” and “differ” to “differed”.
  3. On page 2, third paragraph, move the last sentence to the start of the paragraph.
  4. On page 2, under Material and Methods, change: “…left bundle branch block in ECG according to Strauss criteria…” to “…left bundle branch block (according to Strauss criteria) on ECG.
  5. On page 3, please clarify the meaning of “EF LK”. It seems like “EF” alone would suffice.
  6. The explanation you provided of equal equidistant is very helpful. Reword the last sentence of section 2.2 to incorporate your explanation as follows: “The resampling procedures were performed to obtain equal equidistant resampled time series (time series with the same number of equidistant samples) of RR intervals and respiratory signals [15]”.
  7. I am not a statistician, but it seems to me that the table 2 p definition should be: significance test result for intergroup comparisons (responders vs. non-responders).
  8. In figure 4, please label the blue and green components.
  9. On page 10, second paragraph, change: “Also, it is likely that in these patients, at least in part, there will be a reduction in respiratory muscle weakness” to “In CRT responders, it is also likely (at least in part) that there will be a reduction in respiratory muscle weakness”.
  10. On page 11, third paragraph, you have written: “Therefore, this difference in α1(RR) between responders and non-responders is primarily due to a further decrease in α1(RR) in patients in whom the deterioration of cardiac function was not stopped, but an increase in the value of this parameter in CRT responders”. This is confusing. Did you mean to write: “Therefore, this difference in α1(RR) between responders and non-responders is primarily due to a further decrease in α1(RR) in patients in whom the deterioration of cardiac function was not stopped, rather than an increase in the value of this parameter in CRT responders”? If so, please amend this accordingly.
  11. On page 13, second paragraph, change: “The effect of these changes on the examined parameters of cardiac and respiratory signal complexity, primarily beta blockers, which were shown to not only reduce sympathetic activity, but also increase vagal tone, was not examined and this is a limitation of this study” to “The effect of these changes, primarily from beta blockers (which have been shown to not only reduce sympathetic activity, but also increase vagal tone), on the examined parameters of cardiac and respiratory signal complexity, was not analyzed and this is a limitation of our study”.

Major comment:

  1. If you want to emphasize that baseline and control recordings are not synonymous in your manuscript, the best way to do that would be to change the word “control” to “follow-up”.

Reviewer 2 Report

All comments have been adequately addressed.

Author Response

Thank you for all the suggestions and tips that have contributed to improving the quality of this manuscript.